# The In Vitro Pharmacokinetics of Medicinal Plants: A Review

**DOI:** 10.3390/ph18040551

**Published:** 2025-04-09

**Authors:** Pamela Chaves de Jesus, Deise Maria Rego Rodrigues Silva, Pedro Henrique Macedo Moura, Rajiv Gandhi Gopalsamy, Eloia Emanuelly Dias Silva, Marina dos Santos Barreto, Ronaldy Santana Santos, Allec Yuri Santos Martins, Anne Gabriela de Freitas Almeida, Adriana Kelly Santana Corrêa, Lucas Alves da Mota Santana, Govindasamy Hariharan, Adriana Gibara Guimarães, Lysandro Pinto Borges

**Affiliations:** 1Postgraduate Program in Pharmaceutical Sciences, Federal University of Sergipe, São Cristóvão 49100-000, Brazil; 2Division of Phytochemistry and Drug-Design, Department of Biosciences, Rajagiri College of Social Sciences (Autonomous), Kochi 683104, Kerala, India; 3Department of Biology, Federal University of Sergipe, São Cristóvão 49100-000, Brazil; 4Department of Clinical and Toxicological Analyses, School of Pharmaceutical Sciences, University of São Paulo (USP), São Paulo 05508-220, Brazil; ronaldyss19@gmail.com; 5Department of Pharmacy, Federal University of Sergipe, São Cristóvão 49100-000, Brazil; 6Graduate Program in Dentistry, Federal University of Sergipe, São Cristóvão 49100-000, Brazil; 7School of Sciences, Bharata Mata College (Autonomous), Thrikkakara, Kochi 682021, Kerala, India

**Keywords:** pharmacokinetics, medicinal plant, in vitro techniques, drug kinetics

## Abstract

**Background:** This review examines in vitro techniques for characterizing the pharmacokinetics of medicinal plants, focusing on their role in understanding absorption, distribution, metabolism, and excretion (ADME). The diverse bioactive compounds in medicinal plants highlight the need for robust pharmacokinetic evaluations to ensure their safety and efficacy. **Objectives:** The objectives were to identify and analyze in vitro techniques applied to medicinal plants’ pharmacokinetics, addressing a gap in the literature. **Methods:** Studies were included based on predefined eligibility criteria: in vitro pharmacokinetic studies involving medicinal plants, focusing on ADME stages. Ex vivo, in vivo, and in silico studies were excluded, along with reviews. Data were collected from the PubMed, Web of Science, and Scopus databases in June 2024 using Health Sciences Descriptors (DeCS) and their MeSH synonyms. The data extracted included study location, plant species, bioactive compounds, in vitro protocols, and ADME characteristics. **Results:** The review included 33 studies, with most focusing on metabolism (60%), absorption (25%), or a combination of ADME aspects. Techniques like Caco-2 cells, human liver microsomes, and simulated gastric and intestinal fluids were widely used. **Conclusions:** The findings highlight methodological heterogeneity, including variability in extract preparation, compound concentrations, and experimental conditions, which limits the comparability and clinical applicability of results. Key limitations include the lack of standardized protocols and physiological relevance in in vitro models, underscoring the need for multidisciplinary approaches and integration with in vivo studies.

## 1. Introduction

Medicinal plants play a fundamental role as a source of new drugs for the pharmaceutical industry and are especially valuable due to the vast array of bioactive compounds they harbor. Since the dawn of humanity, plants have been essential in traditional medicine, forming the basis of popular treatments which, over the centuries, have proved effective in various cultures around the world [1,2]. Currently, there is a growing need to understand the biological and/or therapeutic functions of bioactive compounds derived from plants, also known as phytochemicals. The main phytochemical classes studied are terpenes, phenolic compounds, alkaloids, nitrogen compounds, etc. These bioactive compounds are widely used to develop new drugs [3]. The plants *Digitalis* and *Papaver somnniferun* have produced active metabolites that are known and still used today, such as digoxin and morphine, respectively. For medicines containing these active ingredients to be used for human health, they must undergo a series of protocols and validations to ensure their safe use, such as pharmacokinetic tests [4].

Pharmacokinetic studies seek to understand drugs’ absorption, distribution, metabolization, and excretion (ADME). Many factors can interfere with these four processes, the main one being the age of the individual. For example, the same drug will act differently in adults compared to children with regard to the ADME stages [5]. Tests that analyze the ADME of drugs are widely used in pharmaceutical laboratories and must be validated to guarantee drug safety. Studies with cytochrome P450 enzymes, Caco-2 cells, and microsomes are examples of the methodologies applied. However, these methodologies vary according to the characteristics of the bioactive compounds, such as solubility and chemical stability. Lipophilic compounds are analyzed in liver metabolism tests, while hydrophilic compounds are evaluated in intestinal absorption models. In addition, degradation at different pH levels may require gastric and intestinal simulations. Therefore, the choice of method must be adapted to each compound to guarantee accurate results [6]. The search for the ideal drug is the purpose of the drug research and development (R&D) process. However, it is difficult to obtain the ideal drug due to the great complexity of human beings and the ethical issues involved in testing them. For this reason, in vitro models are increasingly being used to carry out pharmacokinetic tests with lower costs, greater safety for those taking part in the studies, and the potential to establish equational hypotheses that describe similar behaviors in humans [7,8].

Therefore, given the scarcity of information on the use of in vitro tests to investigate the pharmacokinetic characteristics of medicinal plants and the growing number of studies looking for alternative methods to in vivo, our goal is to search the national and international literature for data, studies, and/or references that will contribute to the development of future research by providing evidence for possible future applications to produce safe drugs. This review aims to evaluate the studies that have applied in vitro methods during pharmacokinetic research. With the results obtained from the ADME tests reported in the studies, we aim to promote dissemination and discuss the possible implications of these methods as an alternative to in vivo models.

## 2. Methods

### 2.1. Question and PICOS Strategy

The question posed in this review is as follows: “What are the techniques used to characterize the pharmacokinetics of medicinal plants investigated by in vitro tests?”. This study was developed considering the statements available in the Preferred Reporting Items for Systematic Reviews and Meta-Analyses (PRISMA) guidelines [9] and registered on the Open Science Framework (OSF) platform under DOI: 10.17605/OSF.IO/PDBH8 [10].

The PICOS (Population, Intervention, Control, Outcomes and Type of Study) strategy was based on the following: P: the scarcity of studies in the area of pharmacokinetics concerning medicinal plants; I: studies with medicinal plants; C: not applicable; O: to foster knowledge of the in vitro techniques applied to the pharmacokinetics of medicinal plants; S: in vitro tests.

### 2.2. Data Sources and Literature Search

Searches were carried out in the PubMed, Web of Science, and Scopus databases in June 2024, using the following search strategy Health Sciences Descriptors (DeCS): “Medicinal plants”, “in vitro Techniques”, “Pharmacokinetics”, and their respective synonyms according to MeSH/PubMed, listed in Appendix A.

### 2.3. Study Selection and Eligibility Criteria

Two independent reviewers (P.C.J. and D.M.R.R) used the Rayyan tool to review the studies obtained through the electronic search). Potentially relevant studies were selected by reading the title and abstract. Agreement between the reviewers was measured using the kappa statistic [10]. Any discrepancies were resolved by consensus or a third reviewer (A.G.G). The studies were subjected to the following eligibility criteria: articles with descriptors and/or synonyms in the title or abstract (population); studies providing information on the pharmacokinetics of medicinal plants (intervention); studies describing the pharmacokinetic characterization protocol (outcomes); in vitro studies (type of study). In addition, no restrictions regarding the year of publication were applied during the selection of the studies. Reviews; book chapters; abstracts from congresses and/or conferences; in vivo, ex vivo, and in silico studies; and studies that did not use medicinal plants and/or derivatives were excluded. In addition, the authors were contacted by email when access to the full text was not available.

### 2.4. Data Extraction and Visualization

The following information was extracted from the studies included in this review: country, author, plant, bioactive compound, phytochemical class, source, concentration and/or dose of the plant, in vitro protocol, organism used, and ADME stage. The Rayyan platform version 2016 was used to select the articles, and Microsoft Excel^®^ version 365 was used to analyze the data extracted. The VOSviewer software (version 1.6.20 for Windows) was used to create and visualize co-occurrence networks of the important terms extracted from the literature search.

## 3. Results

### 3.1. Search Results and Study Characteristics

As for co-occurrence among the studies included in the systematic review, the most prevalent keywords were “cytochrome p450” (9), “human liver microsomes” (5), and “metabolism” (4), as shown in Figure 1. These terms showed a strong link between each other, and most of these studies were published from 2012 onwards, which indicates the growing application of the three Rs (reduce, replace, and refine) in scientific research in recent times [11]. Keywords grouped in the same cluster are generally connected to similar topics, highlighting the thematic proximity between the studies [12].

Electronic searches in the PubMed (n = 569), Web of Science (n = 1543), and Scopus (n = 701) databases identified a total of 2813 studies. Figure 2 gives an overview of the studies found in the literature search process. Of the total of 2813 studies, 1077 were discarded after duplicates were identified, leaving 1736 articles to be analyzed according to the inclusion criteria. Of these, 1701 studies were excluded because they did not fit the inclusion criteria. Finally, 35 studies following the inclusion criteria were eligible for full reading. Of these, 2 were excluded, 1 because it was duplicated and the other because it was an ex vivo study. In short, 33 studies were included in this systematic review. The kappa test revealed a moderate agreement level (κ = 0.4696) between the reviewers.

### 3.2. Data Analysis

Among the articles that analyzed pharmacokinetics in in vitro technology, we obtained 33 articles [13,14,15,16,17,18,19,20,21,22,23,24,25,26,27,28,29,30,31,32,33,34,35,36,37,38,39,40,41,42,43,44,45]. Of these, 8 (25%) dealt with absorption, 20 (60%) with metabolism, 4 (12%) dealt with absorption and metabolism for the same plant, and only 1 (3%) analyzed the metabolism and excretion of the plant product.

### 3.3. Natural Products and How to Prepare Them

To obtain the natural product for pharmacokinetic analysis, the authors employed various methods, ranging between the use of purified forms used in previous research and that of products already on the market. The present study reveals that China, situated on the Asian continent, had the highest number of published studies involving the utilization of products and in vitro techniques for pharmacokinetics analysis, with fifteen (45.45%) of the studies identified. Subsequently, South Africa was identified as the second-most prevalent country, with four studies (12.12%), followed by South Korea with three (9.09%). Germany, Brazil, Canada, and the United States were also represented with two studies (6.06%) each. The remaining studies were conducted in Finland, Switzerland, Australia, Tunisia, India, and Thailand, collectively accounting for 3.03% of the total. Given that the majority of studies aimed to compare the natural product with the synthetic product available on the market and their potential interaction, some of the articles included utilized an alternative form for analysis, as illustrated below in Table 1 (attached).

### 3.4. Pharmacokinetic Analysis

The results revealed that in vitro techniques for assessing pharmacokinetics were mainly used to analyze metabolism. However, techniques related to absorption and excretion models were also identified. Table 2 (attached) illustrates the category, the technique applied, and the corresponding author. It is worth noting that some studies used more than one organism for analysis, as some aimed to compare the levels of effectiveness, resulting in a total of nineteen organisms being used for pharmacokinetic analysis.

## 4. Discussion

The studies included in this review highlight the growing interest in medicinal plant research over the last few decades, reflecting scientific trends and regional demands. Time-wise, the studies date from 2004, such as that by Lefebvre et al. [13] in Canada, to recent publications, such as that by Grafakou et al. [45] in 2024 in Greece [46]. This period reflects the progressive diversification of extraction techniques, the identification of metabolites, and the preparation of bioactive compounds. Between 2000 and 2010, initial studies focused on the extraction of specific predominate compounds, such as that by Zhao et al. (2011) [16] in China, which investigated β-carboline alkaloids [47]. Between 2011 and 2020, there was a broad expansion in standardized approaches, as seen in the work of Reid et al. (2019) [36] in Finland and Zhou et al. (2020) [39] in China, who explored phenylethanoids and pentacyclic triterpenes, respectively. In the period from 2021 to 2024, studies have focused on both the use of traditional plants and the application of new purification and formulation methodologies, such as the work by Grafakou et al. (2024) [45] with *Tanacetum parthenium* [48].

Geographically, the data from the studies analyzed covered different continents, with greater representation in Asia, particularly in China and South Korea. China stood out due to studies such as those by Fong et al. (2012) [19] and Chang et al. (2022) [42], while South Korea analyzed compounds isolated from plants such as *Magnolia* spp. (Jeong et al., 2013) [22] (Wang et al., 2023) [49]. In Europe, studies carried out in Germany and Greece have brought modern approaches focusing on the laboratory purification of compounds such as alkaloids and sesquiterpene lactones. In the Americas, Brazil stood out with studies such as that by Cruz et al. (2018) [33] on diterpenes and triterpenes in *Euphorbia umbellata*, while in the United States, Fasinu et al. (2017) [30] investigated methanolic extracts of various plants, highlighting the potential of phenolic compounds [50]. In Africa, South African research such as that of Havenga et al. (2018) [32] and Müller et al. (2012) [20] explored local biodiversity with plants such as *Hypoxis hemerocallidea*. Finally, although it is less represented, studies in Oceania, such as that by Bendikov et al. (2017) [31] in Australia, have investigated clerodane diterpenoids [51]. The reviewed studies highlight diverse methodologies for investigating the in vitro pharmacokinetics of medicinal plants, focusing on ADME aspects to predict their safety and efficacy in drug development [52]. The studies use both crude extracts, like *Sutherlandia frutescens* (Müller et al., 2012) [20] and *Terminalia arjuna* (Varghese et al., 2018) [35], reflecting traditional use, and isolated compounds, like *Scutellaria baicalensis* flavonoids (Fong et al., 2012) [19] and *Peganum harmala* alkaloids (Zhao et al., 2011) [16], enabling targeted bioactive analysis [53]. The use of commercial forms, as seen in studies with *Corydalis tubers* (Ji et al., 2011) [17] and *Lycium barbarum* (Feng et al., 2020) [41], impacts pharmacokinetic characterization, limiting reproducibility due to variability in plant material quality and origin [54]. Crude extracts, like methanolic extracts of African plants (Thomfor et al., 2016) [29] and *Sutherlandia frutescens* (Müller et al., 2012) [20], highlight compound synergy but complicate the identification of specific metabolites due to their chemical complexity [55]. Isolated compounds, like *Dodonaea polyandra* diterpenoids (Bendikov et al., 2017) [31], enable detailed pharmacokinetic studies, which are essential for linking chemical structures to pharmacokinetic behavior [56]. Concentrations ranged from 125 nM (pyrrolizidine alkaloids, Kolrep et al., 2018 [34]) to 100 mg/mL (ethanolic extracts, Sumsakul et al., 2015 [25]), reflecting diverse in vitro approaches but questioning biological and translational relevance [57]. Extremely high concentrations may lack physiological relevance, while very low ones can underestimate effects. Broad scales (e.g., 0.01–1000 μM, Ji et al., 2011) address this, but hinder study comparisons due to limited standardization [58]. The analyzed studies reveal a wide range of in vitro pharmacokinetic approaches, covering absorption, metabolism, and distribution, with varied models that present advances and limitations [59]. The use of Caco-2 cells, a standard for simulating the intestinal barrier and predicting the permeability of compounds, was highlighted in seven studies, as evidenced by Havenga et al. (2018) [32] and Mauro et al. (2019) [38,60]. Although simulated gastric and intestinal fluids (SGF and SIF) have been applied in isolated studies, such as those by Cruz et al. (2018) [33] and Chang et al. (2022) [42], their limited application hinders broader validations [61]. Alternative methods, such as PAMPA, used by Petit et al. (2016) [28], offer a simplified analysis of passive permeability, but lack the biological complexity of cell models such as Caco-2 [62]. Metabolism studies are widely explored in 31 papers, 16 of which use human liver microsomes, allowing for the evaluation of phases I and II of metabolism, as well as enzyme inhibition tests [63]. Works such as those by Sun et al. (2010) [15] and Zhao et al. (2011) [16] have investigated the interactions of natural compounds with metabolic enzymes, highlighting the importance of predicting drug interactions [64]. These studies indicate the selective inhibition of cytochrome P450 (CYP) enzymes, such as CYP3A4, CYP2C19, and CYP2C9, reinforcing the need to evaluate interactions between medicinal plants and drugs to avoid adverse effects [65]. Despite the predominance of human microsomes, rat liver microsomes have been used in five studies as a cost-effective alternative, but with results that should be analyzed with caution due to interspecies differences in enzymatic metabolism [66]. Other, less frequent approaches include the use of baculovirus systems in insect cells (Thomfor et al., 2016) [29] and hepatic and intestinal S9 fractions (Kolrep et al., 2018 [34]; Fong et al., 2012 [19]), which allow specific analysis, although they are limited by cost and availability [67]. Distribution, analyzed directly in only one study (Lefebvre et al., 2004) [13], was assessed by fluorimetric microplates to verify the binding of compounds to plasma proteins. The lack of replications limits the applicability of these data in broader contexts [68]. In addition, the use of specialized cell lines, such as HepaRG and C2BBe1, reflects the growing interest in models that mimic human physiology. Although they offer greater predictability for in vivo pharmacokinetics, their high cost and complexity restrict widespread adoption [69]. Absorption has been extensively investigated with methods such as SGF/SIF and Caco-2, while metabolism, with an emphasis on CYP and UGT enzymes, has shown relevant advances in predicting drug interactions. However, gaps remain, particularly in distribution analysis, which needs more replicable studies for validation [70,71]. Figure 3 shows the analysis on the general information, including a temporal analysis, plant samples and country distribution.

The limitations observed in the included studies are diverse, covering both methodological and experimental aspects. In many cases, studies using in vitro models, even with strict control of the experiments, do not fully reproduce the complexity of biological systems in vivo, limiting the extrapolation of results to clinical applications [72]. Furthermore, in vitro analyses, such as those determining enzyme inhibition or evaluating drug–herb interactions, are often conducted with concentrations of compounds that may not reflect physiologically relevant levels, which may not capture inter-individual variability, especially in humans [73,74]. Likewise, many studies lack robust standardization in the techniques for obtaining and preparing extracts or bioactive compounds, which makes reproducibility and comparison between different studies difficult [75].

This systematic review presents some methodological limitations, including the heterogeneity of the experimental methods and the lack of standardization of the protocols used for obtaining extracts and bioactive compounds. A risk of bias assessment was not carried out due to the exploratory and methodological nature of the included studies, which focused on in vitro techniques for pharmacokinetic characterization of medicinal plants. As the results were intended for the laboratory analysis of ADME rather than clinical interventions, the use of non-physiological concentrations and limited data on reproducibility further hinder the translation of in vitro findings to clinical settings. Multidisciplinary approaches, such as the use of molecular biology and computational modeling, can broaden the pharmacokinetic understanding of compounds, while the assessment of inter-individual variability is crucial for the development of personalized and safe drugs, strengthening the use of medicinal plants and their derivatives in the research and development of new drugs.

## 5. Conclusions

This review identified a variety of in vitro techniques used to characterize the pharmacokinetics of medicinal plants, focusing on absorption, metabolism, and, to a lesser extent, distribution and excretion. Commonly employed models include Caco-2 cells for simulating intestinal absorption, human liver microsomes for evaluating enzymatic metabolism and inhibition, and simulated gastric and intestinal fluids (SGF/SIF) for studying compound stability under digestive conditions. Other approaches, such as PAMPA and fluorometric microplates, were used for permeability and protein binding analysis, respectively. Despite their potential, these techniques often lack standardization, and their clinical translation is hindered by methodological variability. Future research should aim to integrate in vitro and in vivo methodologies, establish standardized protocols, and incorporate multidisciplinary approaches to enhance the pharmacokinetic understanding of medicinal plants and their bioactive compounds.

## Figures and Tables

**Figure 1 pharmaceuticals-18-00551-f001:**
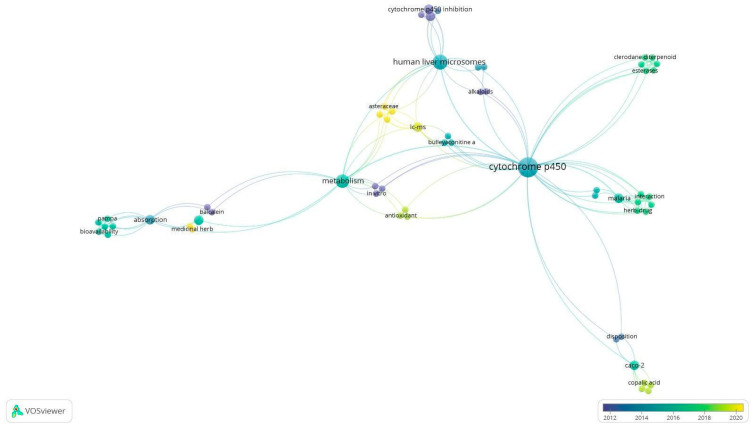
A network map of the co-occurrence in the included studies.

**Figure 2 pharmaceuticals-18-00551-f002:**
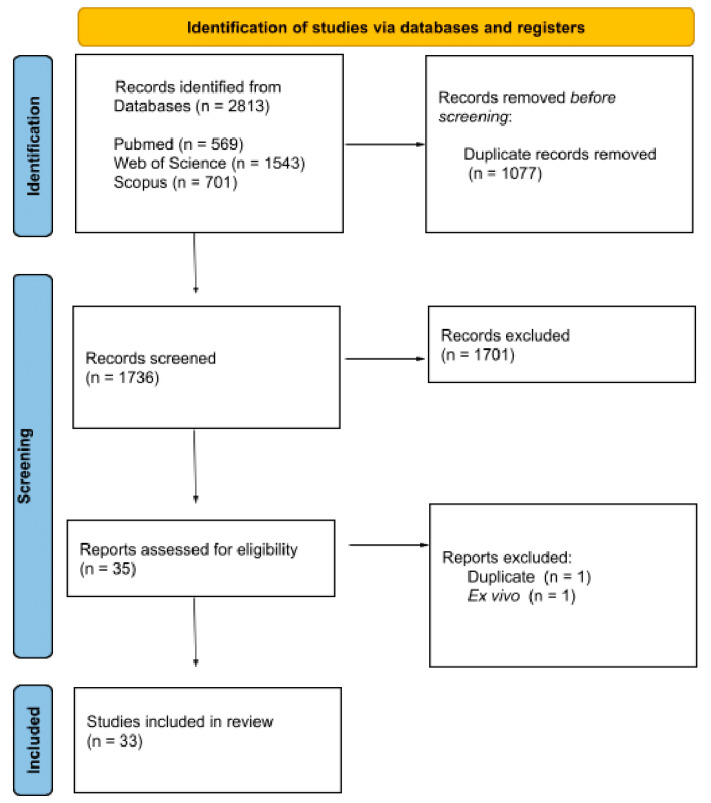
A description of the structure of the results obtained from the search through the aforementioned search platforms. Source: PRISMA, 2020.

**Figure 3 pharmaceuticals-18-00551-f003:**
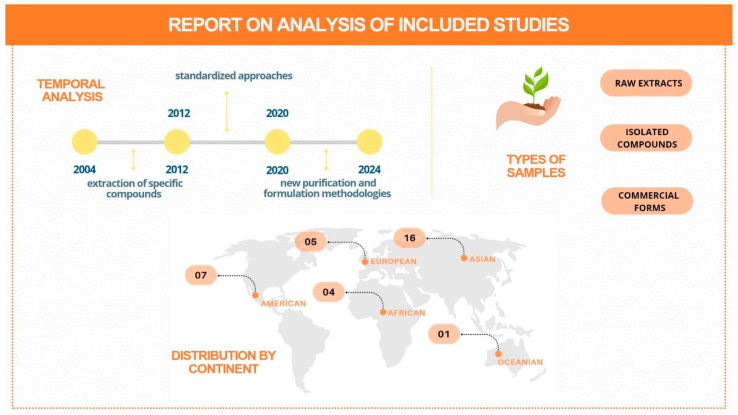
An analysis report on the general information recorded.

**Table 1 pharmaceuticals-18-00551-t001:** General characteristics of the 33 studies included.

Author	Country	Plant	Metabolite	Form(s) of Preparation	Concentration or Dose
Lefebvre et al., 2004 [13]	Canada	* Valeriana officinalis * L.	Vairidoids, Monoterpenes And Sesquiterpenes	Commercial Product	Capsules, tablets and caplets were extracted at a concentration of 100 mg/mL. Teas were extracted at a concentration of 25 mg/mL
Lee et al., 2008 [14]	South Korea	* Anthriscus sylvestris *	Lignans	Were Isolated/Purified In The Laboratory Of The Research Institution	50 µM
Sun et al., 2010 [15]	China	**	Flavonoid	Were Isolated/Purified In The Laboratory Of The Research Institution	various concentrations
Zhao et al., 2011 [16]	China	* Peganum harmala *	Β-Carboline Alkaloids	Solutions Of Alkaloids	various concentrations
Ji et al., 2011 [17]	China	* Aconitum bulleyanum *	diester-diterpene alkaloid	Commercial Product	50 μmol
Han et al., 2011 [18]	China	Huanglian (*Rhizoma coptidis*)	Alkaloids	Hot Water Extracts	various concentrations
Fong et al., 2012 [19]	China	* Scutellaria baicalensis * Georgi	Flavone Isolated	Solutions	final concentration of 5 μM
Müller et al., 2012 [20]	South Africa	* Sutherlandia Frutescens *	Triterpenoid Glycosides, Glycosides Of Flavonols, Non-Protein Amino Acid.	Aqueous And Methanolic Extracts	10 mg/mL
					10 mg/mL
Cieniak et al., 2013 [21]	Canada	Cree plants-list of 17 species	**	Ethanolic And Methanolic Extracts	100 μg/mL
					50 μg/mL
					5 mg/mL
					10 μL
					**
Jeong et al., 2013 [22]	South Korea	* Magnolia officinalis, Magnolia grandiflora * and other plants	**	Commercial Product	Various Concentrations of honokiol (0.05–100 μM)
					Various Concentrations of honokiol (1–200 μM for UGT1A1, UGT1A4, and UGT2B7; 0.01–2 μM for UGT1A9)
					Various Concentrations
Bi et al., 2013 [23]	South Korea	* Corydalis tubers *	Alkaloid	Commercial Product	Final Concentrations of 1–200 μM
					Final Concentrations of 1–1000 μM
					Various Concentrations
Feng et al., 2014 [24]	China	* Halenia elliptica * D. Don	**	Were Isolated/Purified In The Laboratory Of The Research Institution	1–500 μM
Sumsakul et al., 2015 [25]	Thailand	* Plumbago indica * Linn., *Garcinia mangostana* Linn., *Dracaena loureiri* Gagnepv, *Dioscorea membranacea* Pierre and *Myristica fragans* Houtt.	**	Ethanolic Extracts	100 mg/mL
Khadhri et al., 2015 [26]	Tunisia	* Ruta chalepensis * L. and *Ruta montana* L.	Polyphenol	Ethanol Extracts	4 mg/mL
Kan et al., 2015 [27]	China	Wu-tou decoction (Q) is composed of *Aconiti Radix Cocta, Ephedrae Herba, Paeoniae Radix Alba, Astragali Radix* and *Glycyrrhiza Radix Preparata*	Alkaloids	Commercial Product	various concentrations
Petit C et al., 2016 [28]	Switzerland	* Angelica archangelica * (L.) H.Karst.	Furanocoumarins	Methanolic Extracts	10 mg/mL
		* Waltheria indica * L.	Alkaloids	Aqueous Extract	
		* Pueraria montana * var. lobata (Willd.) Sanjappa & Pradeep	Flavonoids	Hot Water Extracts	
Thomfor et al., 2016 [29]	South Africa	*Hyptis suaveolens*, *Boerhavia diffusa*, *Newbouldia laevis*, *Launaea taraxacifolia* and *Myrothamnus flabellifolius*	**	Aqueous Extracts	10 µg/mL and 100 µg/mL
Fasinu et al., 2017 [30]	United States	*Annona muricata*, *Argermone mexicana*, *Kalanchoe pinnata*, *Mangifera indica*, *Momordica charantia*, *Phyllanthus amarus and Tithonia diversifolia.*	Alkaloids, Nematicidal Compounds, Phenolics	Methanolic Extracts	2 µg/mL
Bendikov et al., 2017 [31]	Australia	* Dodonaea polyandra *	Clerodane Diterpenoids	Were Isolated/Purified In The Laboratory Of The Research Institution	10 μM
Havenga et al., 2018 [32]	South Africa	* Hypoxis hemerocallidea *	**	Reference Dried Plant Material	500 µg/mL
				Commercial Product	
				Aqueous Extract	
Cruz et al., 2018 [33]	Brazil	* Euphorbia umbellata * (Pax) Bruyns	Diterpenes And Triterpenes	Suspension With Sulphoric Acid	100 µM
					1.17 mM
Kolrep et al., 2018 [34]	Germany	**	Pyrrolizidine alkaloids	Commercial Product	125 nM PA.
Varghese et al., 2018 [35]	India	* Terminalia arjuna *	**	Methanolic Extract	various concentrations
Reid et al., 2019 [36]	Finland	* Lippia scaberrima *	Phenylethanoid	Commercial Product	10 µM
					0.4, 2, 10, and 50 µM
Bräuer et al., 2019 [37]	Germany	* Eurycoma longifolia *	Triterpenes, Alkaloids	Extracts	final concentration of 10 μM
Mauro et al., 2019 [38]	Brazil	* Copaifera langsdorffii *	Diterpenes	Commercial Product	**
Zhou et al., 2020 [39]	China	* Pulsatilla chinensis *	Pentacyclic Triterpene	Were Isolated/Purified In The Laboratory Of The Research Institution	0.5, 1, 2, 5, 10, 20, 50, 100 μM
					1 µM
Wang et al., 2020 [40]	China	**	Flavonoids	Commercial Product	Various Concentrations
Feng et al., 2020 [41]	China	* Lycium barbarum * L.	Polysaccharides	Ethanol Extracts	**
					various concentrations
					200 μg/mL
Chang et al., 2022 [42]	China	* Lonicera japonica * Flos	Flavones, Organic Acids And Iridoids	Aqueous Extract	1 mL
Kane et al., 2022 [43]	United States	* Artemisia annua * L. cv. SAM (voucher MASS 317314)	Flavonoids	Hot Water Extracts	various concentrations
		* Artemisia afra * SEN (voucher LG0019529)			
		* Artemisia afra * MAL (voucher FTG, 181107)			
Husain et al., 2022 [44]	United States	* Bulbine natalensis * Baker	**	Methanolic Extract	various concentrations
Grafakou et al., 2024 [45]	Greence	* Tanacetum *	Sesquiterpene Lactones	Were Isolated/Purified During Previous Work-University Laboratory	10 μM test compound
		* parthenium *			
		* Cynara * spp.,			
		* Crepis * spp			
		* Calea * spp.			
		* Centaurea * spp.			
		* Artemisia *			
		* dubia, Achillea *			
		* coarctata *			

****** No information attributed by the authors.

**Table 2 pharmaceuticals-18-00551-t002:** Representation of in vitro techniques used for pharmacokinetic evaluation of herbal products.

Technique/System	N° of Studies	Category	References
Caco-2 cells	7	Absorption	Havenga et al., 2018 [32]; Fong et al., 2012 [19]; Müller et al., 2012 [20]; Wang et al., 2020 [40]; Feng et al., 2020 [41]; Mauro et al., 2019 [38]; Kan et al., 2015 [27]
Simulated gastric juice	1	Absorption	Chang et al., 2022 [42]
Simulated gastric (SGF) and intestinal fluids (SIF)	1	Absorption	Cruz et al., 2018 [33]
Synthetic gastric and pancreatic juices	1	Absorption	Khadhri et al., 2015 [26]
C2BBe1 cell line	1	Absorption	Cieniak et al., 2013 [21]
PAMPA (Parallel Artificial Membrane Permeability Assay)	1	Absorption	Petit et al., 2016 [28]
Human liver microsomes-Incubation (phase I, II, or combined)	1	Metabolism	Grafakou et al., 2024 [45]
Human liver microsomes-Enzyme inhibition	15	Metabolism	Sun et al., 2010 [15]; Zhao et al., 2011 [16]; Zhou et al., 2020 [39]; Sumsakul et al., 2015 [25]; Bendikov et al., 2017 [31]; Jeong et al., 2013 [22]; Ji et al., 2013 [17]; Han et al., 2011 [18]; Kane et al., 2022 [43]; Cruz et al., 2018 [33]
UDGPA	1	Metabolism	Cieniak et al., 2013 [21]
Rat liver microsomes	4	Metabolism	Bräuer et al., 2019 [37]; Lee et al., 2008 [14]; Bi et al., 2011 [23]; Varghese et al., 2018 [35]
Placental microsomes	1	Metabolism	Reid et al., 2019 [36]
Rat intestinal S9 (RIs9)	1	Metabolism	Fong et al., 2012 [19]
CYP proteins	1	Metabolism	Fasinu et al., 2017 [30]
hMDR1-MDCKII cells	1	Metabolism	Fasinu et al., 2017 [30]
Baculovirus system in insect cells	1	Metabolism	Thomfor et al., 2016 [29]
HepaRG cells	2	Metabolism	Kane et al., 2022 [43]; Husain et al., 2022 [44]
Liver S9 fractions	1	Metabolism	Kolrep et al., 2018 [34]
Fluorometric microplates	1	Metabolism	Lefebvre et al., 2004 [13]
Fluorometric microplates	1	Distribution	Lefebvre et al., 2004 [13]

## Data Availability

Not applicable.

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
