# Peer review of "The In Vitro Pharmacokinetics of Medicinal Plants: A Review"

_pharmaceuticals, 2025, doi:10.3390/ph18040551_

Round 1

Reviewer 1 Report

Comments and Suggestions for Authors

The manuscript "In Vitro Pharmacokinetics of Medicinal Plants: A Review" provides a systemic analysis of various pharmacokinetic evaluation techniques. The manuscript need substantial revision before acceptance. 

1. Abstract: "The review is registered on the Open Science Framework (DOI: 10.17605/OSF.IO/PDBH8) and received no specific funding." It should be moved to acknowledgment or conflict of interest rather than adding to the abstract. 

2. Introduction: A paragraph on the novelty of the work and major findings should be included at the end of the introduction. 

3. There is no numbering system for headings, although subheading numbers are given. Use a uniform pattern for the numbering of headings and subheadings in the manuscript. 

4. Methods section 2.1: "DOI: 10.17605/OSF.IO/PD10BH8" Check the citation carefully. The reference is cited in between the DOI number. 

5. "Of the total of 2813 studies, 1745 were discarded after duplicates were identified, leaving 1736 articles to be analyzed..." The statement is confusing or miscalculating. Need further explanation for the statement. 

6. "Of these, 1701 studies were excluded." Discuss the reason for the exclusion of these 1701 studies and the selection of only 35 studies along with this statement. 

7.  "Of these, two were excluded, one because it was duplicated...." The duplications were already removed already in the first step. The authors need to correct their methodology and revise the manuscript. 

8. A network map for the in vitro techniques also should be added for different techniques of pharmacokinetics. 

9. Figure 2 is unreadable. 

10. Use uniform pattern (Italic) for words "In vitro, in vivo, in silico, etc..)

11. Revise the manuscript thoroughly for typos and grammatical errors. 

Author Response

INDEPENDENT REVIEW REPORT, REVIEWER 1 (ROUND 1)

1. Abstract: "The review is registered on the Open Science Framework (DOI: 10.17605/OSF.IO/PDBH8) and received no specific funding." It should be moved to acknowledgement or conflict of interest rather than adding to the abstract.

We have removed this information from the abstract and added it to the Funding section.

2. Introduction: A paragraph on the novelty of the work and major findings should be included at the end of the introduction.

We have added the following passage to the introduction:

“This review aims to evaluate the studies that have applied in vitro methods for pharmacokinetic research. With the results obtained from the ADME tests reported in the studies, we aim to promote the dissemination and possible implications of these methods as an alternative to using in vivo models.”

3. There is no numbering system for headings, although subheading numbers are given. Use a uniform pattern for the numbering of headings and subheadings in the manuscript.

The numbering of the headings and subheadings has been added to the manuscript.

4. Methods section 2.1: "DOI: 10.17605/OSF.IO/PD10BH8" Check the citation carefully. The reference is cited in between the DOI number.

We apologize for the error. We have added the reference after the DOI number.

5. "Of the total of 2813 studies, 1745 were discarded after duplicates were identified, leaving 1736 articles to be analyzed..." The statement is confusing or miscalculating. Need further explanation for the statement.

The count is in, here's the text:

Of the total of 2813 studies, 1077 were discarded after duplicates were identified, leaving 1736 articles to be analyzed according to the inclusion criteria.

6. "Of these, 1701 studies were excluded." Discuss the reason for the exclusion of these 1701 studies and the selection of only 35 studies along with this statement.

We have added this information to the statement:

Of these, 1077 studies were excluded because they did not fit the inclusion criteria. Finally, 35 studies met the inclusion criteria and were eligible for full reading.

The review's eligibility criteria are listed in section 2.3 of the methodology.

7. "Of these, two were excluded, one because it was duplicated...." The duplications were already removed in the first step. The authors need to correct their methodology and revise the manuscript.

In this case, these studies were not initially identified as duplicates on the Rayyan platform, since the titles were in different languages. Therefore, they went through the title and abstract selection, and were excluded after reading and detecting the duplicate. Since they went through the initial reading process, we added the exclusion after this process.

8. A network map for the in vitro techniques also should be added for different techniques of pharmacokinetics.

At this point, I would appreciate it if you could explain to me what you want. Would it be another VosViewer, with data from the main techniques applied in vitro? Or could it be another one? I'm looking forward to your feedback on this point in order to improve it.

9. Figure 2 is unreadable.

Done

10. Use a uniform pattern (Italic) for words "In vitro, in vivo, in silico, etc..)

Quoted words have been revised throughout the manuscript and added in italics.

11. Revise the manuscript thoroughly for typos and grammatical errors.

The manuscript has been revised entirely to correct grammatical errors.

Reviewer 2 Report

Comments and Suggestions for Authors

The submitted publication features a promising title and abstract, but the content does not fully align with expectations. The study of in vitro pharmacokinetics of medicinal plants is a crucial step in drug research and development. This area of investigation helps us understand how active compounds derived from plants interact with biological systems at the molecular and cellular levels, without the need for animal testing, thereby adhering to the 3R principles (Replacement, Reduction, Refinement). As such, the relevance of this topic is undeniable. However, the authors have limited their work to a superficial analysis of the prevalence of in vitro pharmacokinetic techniques, which is insufficient for a comprehensive evaluation.

To enhance the manuscript, it is essential to include a detailed description of the methods used and their significance in in vitro pharmacokinetic studies. Additionally, the authors should identify the minimum set of techniques required to adequately describe the pharmacokinetics of medicinal plants. Are there differences in the methodologies applied to various types of compounds? Addressing these questions would significantly improve the quality of the manuscript and increase its scientific value and appeal.

Author Response

To enhance the manuscript, it is essential to include a detailed description of the methods used and their significance in in vitro pharmacokinetic studies. Additionally, the authors should identify the minimum set of techniques required to adequately describe the pharmacokinetics of medicinal plants. Are there differences in the methodologies applied to various types of compounds? Addressing these questions would significantly improve the quality of the manuscript and increase its scientific value and appeal.

Thank you for your suggestions and guidance. More information about this question has been inserted in the introduction section. I look forward to any further suggestions for improvement.

It follows in the text:

“Tests that analyze the ADME of drugs are widely used in pharmaceutical laboratories and must be validated to guarantee drug safety. Studies with cytochrome P450 enzymes, Caco-2 cells and microsomes are examples of methodologies applied. However, these methodologies vary according to the characteristics of the bioactive compounds, such as solubility and chemical stability. Lipophilic compounds are analyzed in liver metabolism tests, while hydrophilic compounds are evaluated in intestinal absorption models. In addition, degradation at different pH levels may require gastric and intestinal simulations. Therefore, the choice of method must be adapted to each compound to guarantee accurate results.”

Reviewer 3 Report

Comments and Suggestions for Authors

The manuscript by Borges et al. describes a bibliographical analysis of the literature describing  in vitro studies of pharmokinetic of medicinal plants. It is concluded that this approach as a stand alone method is not sufficiently reliable and must be combined with in vivo studies to improve  understanding of the pharmacokinetic of medicinal plants and their bioactive compounds.

The paper consists of two parts. The first part describes the approach for the analysis of literature, whereas the results of the literature analysis are presented in the second part. Noteworthy, the first part is written in much detail with inclusion of two diagrams of purely technical  character. In contrast, the second pert that must describe the reults interesting for specialists is written as a solid continuous text and is very hardly readable.

I think that this approach is wrong and inconvenient for the reader. I suggest that the authors present the results of their research in the form of easily understandable tables and/or diagrams, whereas the diagrams in the first part can be easily omitted. 

I recommend reconsideration after major revision.

Author Response

The paper consists of two parts. The first part describes the approach for the analysis of literature, whereas the results of the literature analysis are presented in the second part. Noteworthy, the first part is written in much detail with inclusion of two diagrams of purely technical character. In contrast, the second pert that must describe the reults interesting for specialists is written as a solid continuous text and is very hardly readable.

I think that this approach is wrong and inconvenient for the reader. I suggest that the authors present the results of their research in the form of easily understandable tables and/or diagrams, whereas the diagrams in the first part can be easily omitted.

Thank you for your suggestions and guidance. A figure containing more summarized information on the overall results has been added to the manuscript. I look forward to any further suggestions for improvement.

Figure 3 follows:

“Figure 3 - Analysis report on general information.”

Round 2

Reviewer 2 Report

Comments and Suggestions for Authors

Some changes have been made to the manuscript to improve its quality so that it can be published. 

Reviewer 3 Report

Comments and Suggestions for Authors

The authors sufficiently improved the manuscript